# CRAFTING DATA-FREE UNIVERSAL ADVERSARIES WITH DILATE LOSS

## ABSTRACT

We introduce a method to create Universal Adversarial Perturbations (UAP) for a given CNN in a data-free manner. Data-free approaches suite scenarios where the original training data is unavailable for crafting adversaries. We show that the adversary generation with full training data can be approximated to a formulation without data. This is realized through a sequential optimization of the adversarial perturbation with the proposed *dilate loss*. *Dilate loss* basically maximizes the Euclidean norm of the output before nonlinearity at any layer. By doing so, the perturbation constrains the ReLU activation function at every layer to act roughly linear for data points and thus eliminate the dependency on data for crafting UAPs. Extensive experiments demonstrate that our method not only has theoretical support, but achieves higher fooling rate than the existing data-free work. Furthermore, we evidence improvement in limited data cases as well.

## 1 INTRODUCTION

Despite the phenomenal success of deep neural networks in many practical applications, adversarial attacks are being a constant plague. These attacks corrupt the input with a small and usually imperceptible structured noise causing the model to output incorrect predictions. The sole existence of such a vulnerability not only raises concerns about the security of deep learning models, but also questions the robustness of the learned representations. To make it further worse, it has been shown that a single noise, called universal adversarial perturbation (UAP), can be added to any image and fool the network. UAPs do not require any optimization on the input image at attack time, but the corruption effectively works for most of the images. Interestingly, such perturbations created for one model exhibit transferability of attack and induce high fooling on other models. One drawback of UAPs though, is the requirement of training data for crafting perturbations. This is increasingly infeasible as the datasets are becoming quite large and might not be publicly released due to privacy or copyright reasons. In such cases where the original data is not available, *data-free* methods are gaining traction. In the data-free setting, the perturbation is created only with the trained neural network. Such methods typically rely on the trained weights and the CNN structure to find vulnerable patterns that can maximally disturb the normal propagation of activations across the network. A higher transfer of attack across networks is observed for data-free UAPs as well, raising its practical utility. Moreover, the study of these perturbations might lead to new insights on how deep neural networks actually work.

In this paper, we propose a new method for crafting data-free UAPs for any given CNN using ReLU nonlinearity. The approach relies on finding the singular vectors of a linearly approximated network (Section 3.1). A loss formulation is devised to enable this approximation under certain conditions. *Dilate loss* forms the major component of the method, which generates a perturbation that maximizes the Euclidean norm of the activation vector (before the nonlinearity) at a given layer (Section 3.2). We show that the perturbation crafted through *dilation* has the effect of linearly approximating the ReLU layer responses for any data points. These *dilations* are done sequentially for all the layers from the input to the last classification stage (Section 3.3). We argue that the *sequential dilations* results in a perturbation that aligns with the first singular vector of the linearly approximated network. Our approach outperforms the existing data-free method in fooling rates and the evaluation is also done for less data scenarios (Section 4).

In summary, the work contributes the following:

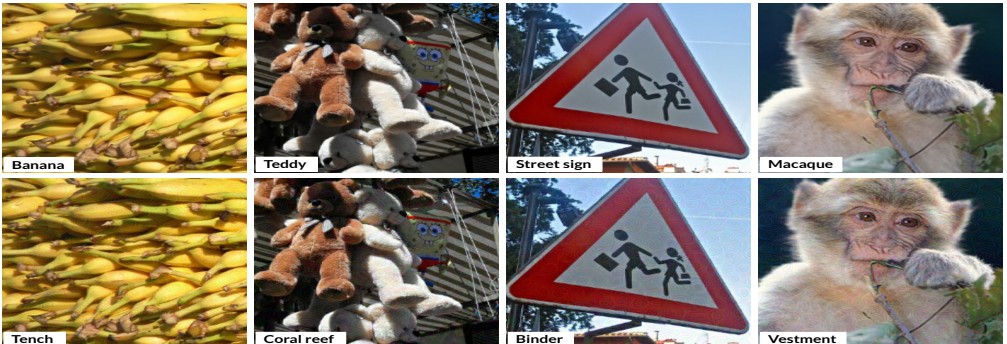

Figure 1: Demonstration of adversarial attack through perturbation generated from our proposed sequential dilation algorithm. The first row shows the clean images with the class predicted by the model, while the second row has the corresponding perturbed samples with the flipped labels.

- A new method that can create universal adversarial perturbation without using data and achieve state-of-the-art data-free fooling rates.
- A detailed theoretical analysis which formulates the proposed sequential dilation algorithm by approximating the adversary generation with full training data under certain conditions.

## 2 RELATED WORK

The vulnerability of deep neural networks to adversarial samples is first shown in Szegedy et al. (2013). Following Szegedy et al. (2013), several methods (Goodfellow et al., 2014; Kurakin et al., 2016; Dong et al., 2018; Madry et al., 2017; Moosavi-Dezfooli et al., 2016; Brendel et al., 2017; Athalye et al., 2018; Carlini & Wagner, 2017) are being proposed to craft such adversarial samples. One of the simplest method is the Fast Gradient Sign Method (FGSM) formulated in Goodfellow et al. (2014). FGSM obtains the perturbation by single step gradient ascent of the loss function with respect to the input image. There are multi step variants to FGSM like iterative FGSM (Kurakin et al., 2016), Momentum (Dong et al., 2018), Projected Gradient Descent (PGD) (Madry et al., 2017), Deepfool (Moosavi-Dezfooli et al., 2016), etc. These attacks are image-specific, where the perturbation is a function of the input and requires a separate optimization for each image.

Moosavi-Dezfooli et al. (2017) introduce the idea of Universal Adversarial Perturbations (UAP), a single perturbation that can fool the model for most of the input images. UAP is obtained by jointly maximizing the training loss for dataset images. There are also generative approaches like NAG (Reddy Mopuri et al., 2018b), AAA (Reddy Mopuri et al., 2018a), GAP (Poursaeed et al., 2018) for crafting universal adversaries. Khrulkov & Oseledets (2018) propose a method based on singular vectors of Jacobian matrix to create universal adversaries. They show impressive fooling performance with a very small set of training images, but the method is not data-free. Though the study of adversarial attacks started with the classification task, there are several works (Xie et al., 2017; Metzen et al., 2017) that extend such attacks to other tasks like segmentation, detection, etc. Further, adversarial examples are shown to generalize to the physical world in Kurakin et al. (2016). While most attacks changes each pixel in the image with small imperceptible noise, there are methods (Sharif et al., 2016; Brown et al., 2017; Papernot et al., 2016) that perturb limited number of pixels with large noise as these are more practical in nature.

The attacks discussed so far, in general, rely on maximizing the training loss. In contrast, Mopuri et al. (2018) devise a generalizable data-free objective for crafting UAPs (GDUAP). GDUAP maximizes the activations at the output of all convolutional layers corrupting the feature representations learned and hence fooling the model. Our method has similarity to GDUAP, but with the crucial difference that the Euclidean norm maximization is performed before the nonlinearity in our case. Further, we maximize the norms of the layers one after the other in a sequential fashion as opposed to a single joint optimization. We show theoretically and experimentally that these changes cause a lot of difference in fooling performance. Moreover, no sound reasoning is available in Mopuri et al. (2018) to justify the formulation, whereas we provide theoretical explanation for the algorithm.

## 3 OUR APPROACH

### 3.1 CRAFTING A DATA-FREE OBJECTIVE

Consider a deep neural network with $L$ layers already trained for classification task. We assume the activation function employed in the network to be ReLU (Nair & Hinton, 2010), defined as

$$\sigma_R(x) = \begin{cases} x & \text{if } x > 0 \\ 0 & \text{otherwise,} \end{cases}$$

which basically zeros out all negative elements and retains the positive ones when applied on a matrix. Let $f_1(\boldsymbol{x}) = \boldsymbol{W}_1\boldsymbol{x}$, $f_2(\boldsymbol{x}) = \boldsymbol{W}_2\sigma_R(\boldsymbol{W}_1\boldsymbol{x})$, ..., $f_l(\boldsymbol{x}) = \boldsymbol{W}_l\sigma_R(\dots \boldsymbol{W}_2\sigma_R(\boldsymbol{W}_1\boldsymbol{x})\dots)$ be the outputs at different layers of the network for an input vector $\boldsymbol{x}$. Note that the output $f_i$ for the $i$th layer is taken before the nonlinearity and $f_L$ represents the pre-softmax neuron layer. We ignore the bias terms for mathematical simplicity. The weights ($\boldsymbol{W}_i$s) of the network are trained with input and label pairs from the dataset $\mathbb{D}$.

Our aim is to craft a perturbation vector $\boldsymbol{p}$ with Euclidean norm $c$ such that the network incorrectly classifies for most of the data samples. Mathematically, the optimization can be written as,

$$\max_{\boldsymbol{p}:|\boldsymbol{p}|=c} \sum_{(\mathbf{x}_i, y_i) \in \mathbb{D}} \mathcal{I}_{\operatorname{argmax}(\sigma_S(f_L(\boldsymbol{x}_i + \boldsymbol{p}))) \neq y_i}, \tag{1}$$

where $\sigma_S$ is the softmax function. Note that the condition for the indicator function ($\mathcal{I}$) that checks for misclassification is dependent on the ground truth labels ($y_i$s). Assuming a high classification accuracy for the model, we approximate the condition to $\operatorname{argmax}(\sigma_S(f_L(\boldsymbol{x}_i + \boldsymbol{p}))) \neq \operatorname{argmax}(\sigma_S(f_L(\boldsymbol{x}_i)))$. Since softmax function is monotonic, we further relax the objective 1 to

$$\max_{\boldsymbol{p}:|\boldsymbol{p}|=c} \sum_{(\boldsymbol{x}_i, y_i) \in \mathbb{D}} |f_L(\boldsymbol{x}_i + \boldsymbol{p}) - f_L(\boldsymbol{x}_i)|^2, \tag{2}$$

which amounts to finding a $\boldsymbol{p}$ that maximizes the network response after being added to the inputs. In other words, $\boldsymbol{p}$ should maximally disturb the output $f_L$ for all data points. Note that for some $\boldsymbol{x}_i$s, maximizing $|f_L(\boldsymbol{x}_i + \boldsymbol{p}) - f_L(\boldsymbol{x}_i)|^2$ might not result in incorrect prediction. We assume such cases to be minority and the objective could lead to significant adversarial changes to $f_L$ responses for majority input samples. If $\boldsymbol{X} = \{\boldsymbol{x}_0, \boldsymbol{x}_1, \dots, \boldsymbol{x}_N\}$ denote the matrix formed by the assembling all the $N$ data samples as columns and $\mathbf{1}$ represent a column vector of ones of appropriate size, then the optimization 2 can be rewritten as,

$$\max_{\boldsymbol{p}:|\boldsymbol{p}|=c} |f_L(\boldsymbol{X} + \boldsymbol{p}\mathbf{1}^T) - f_L(\boldsymbol{X})|_F^2. \tag{3}$$

We recognize that optimization 3 is not exactly equivalent to the original objective 1, but is an approximation which does not require the ground truth labels. But still the training data $\boldsymbol{X}$ is essential for computation and needs to be eliminated for a complete data-free approach. Now observe that if $f_L$ were to be a linear function, then the objective 3 reduces to,

$$\max_{\boldsymbol{p}:|\boldsymbol{p}|=c} |f_L(\boldsymbol{p})|^2, \tag{4}$$

which means that $\boldsymbol{p}$ has to align along the first right singular vector of the linear $f_L$ map. The singular $\boldsymbol{p}$ could potentially disturb the output $f_L$ more for all the $\boldsymbol{x}_i$s than any other vector. Interestingly, note that the optimization 4 is a data-free objective under the linear assumption of $f_L$. However, $f_L$ is nonlinear due to the presence of ReLU activation functions at every layer. Note that the formulation 4 is valid even if $f_L$ is not a complete linear map, but satisfies $f_L(\boldsymbol{X} + \boldsymbol{p}\mathbf{1}^T) = f_L(\boldsymbol{X}) + f_L(\boldsymbol{p}\mathbf{1}^T)$ for some $\boldsymbol{p}$. Hence, we devise an algorithm to seek a perturbation that can approximately induce the above additivity property to the ReLU network.

### 3.2 LINEARLY APPROXIMATING THE NETWORK

We start by noting that the only nonlinearity in the network is due to the ReLU activation function at every layer. But ReLU is piece-wise linear; especially, observe that $\sigma_R(\boldsymbol{a} + \boldsymbol{b}) = \sigma_R(\boldsymbol{a}) + \sigma_R(\boldsymbol{b})$ if vectors $\boldsymbol{a}$ and $\boldsymbol{b}$ are in the same orthant. Now consider the ReLU nonlinearity after the first layer,

$\sigma_R(\boldsymbol{W}_1\boldsymbol{X} + \boldsymbol{W}_1\boldsymbol{p}1^T)$, which becomes additive if column vectors in $\boldsymbol{W}_1\boldsymbol{X}$ are in the same orthant as $\boldsymbol{W}_1\boldsymbol{p}$. We relax this criteria and favour the case of making the vectors as close as possible by,

$$\max_{\boldsymbol{p}:|\boldsymbol{p}|=c} \mathbf{1}^T(\boldsymbol{W}_1\boldsymbol{X})^T(\boldsymbol{W}_1\boldsymbol{p}) = N(\boldsymbol{W}_1\bar{\boldsymbol{x}}_1)^T(\boldsymbol{W}_1\boldsymbol{p}), \tag{5}$$

where $\bar{\boldsymbol{x}}_1$ stands for the mean of the $N$ data samples in $X$. The solution of the optimization 5 is expected to minimize the error due to the additive approximation of the layer. In order to eliminate the data term from the objective, we make an assumption that the first singular vector of the weight matrices align along the mean vector of its corresponding input. In other words, the dot product of data mean $\bar{\boldsymbol{x}}_1$ with the singular vectors of $\boldsymbol{W}_1$ is maximum for the first singular vector. Now we use the following lemma to argue that the objective 5 is maximum when $\boldsymbol{p}$ aligns with the first singular vector of $\boldsymbol{W}_1$ (proof available in Appendix A).

**Lemma 1.** *If $\boldsymbol{x}$ has positive and larger scalar projection on the first singular vector of $\boldsymbol{W}$ than remaining singular vectors, then $argmax_{\boldsymbol{p}}\boldsymbol{x}\boldsymbol{W}^T\boldsymbol{W}\boldsymbol{p} = argmax_{\boldsymbol{p}}|\boldsymbol{W}\boldsymbol{p}|^2$ subject to $|\boldsymbol{p}| = c$.*

Hence, the optimization problem 5 is equivalent to,

$$\max_{\boldsymbol{p}:|\boldsymbol{p}|=c} |\boldsymbol{W}_1\boldsymbol{p}|^2, \tag{6}$$

which we call as the *dilation* of the first layer. We justify the assumption based on the premise that the singular vectors of the weights must have captured the discriminatory modes of data samples while training. By discriminatory mode we refer to the components of $\boldsymbol{X}$ that are essential for the classification task and most likely extracted by the hierarchy of weights in the network. These does not correspond to the modes of variation of data points. The assumption essentially means that the first singular vector carries the most important features common to most of the data points than the remaining singular directions. This is taken to be valid for any layer weight $W_l$ with difference that the mean vector $\bar{\boldsymbol{x}}_l$ is averaged over the layer $l-1$ output, i.e. $\bar{\boldsymbol{x}}_l = (1/N)\sigma_R(f_{l-1}(\boldsymbol{X}))\mathbf{1}$ for $l > 1$.

Now consider the second layer of the network given by $\sigma_R(\boldsymbol{W}_2\sigma_R(\boldsymbol{W}_1\boldsymbol{X} + \boldsymbol{W}_1\boldsymbol{p}1^T))$, where there are two ReLU functions in action. Suppose the first ReLU function is linearly approximated with dilation objective 6. Consequently, the second layer output can be written as $\sigma_R(\boldsymbol{W}_2\sigma_R(\boldsymbol{W}_1\boldsymbol{X}) + \boldsymbol{W}_2\sigma_R(\boldsymbol{W}_1\boldsymbol{p}1^T))$. Note that the second ReLU can be linearly approximated if column vectors in $\boldsymbol{W}_2\sigma_R(\boldsymbol{W}_1\boldsymbol{X})$ are close to $\boldsymbol{W}_2\sigma_R(\boldsymbol{W}_1\boldsymbol{p})$. Considering the two approximations, we formulate the optimization as,

$$\max_{\boldsymbol{p}:|\boldsymbol{p}|=c} \mathbf{1}^T(\boldsymbol{W}_2\sigma_R(\boldsymbol{W}_1\boldsymbol{X}))^T(\boldsymbol{W}_2\sigma_R(\boldsymbol{W}_1\boldsymbol{p})) + \mathbf{1}^T(\boldsymbol{W}_1\boldsymbol{X})^T(\boldsymbol{W}_1\boldsymbol{p}), \tag{7}$$

$$\max_{\boldsymbol{p}:|\boldsymbol{p}|=c} (\boldsymbol{W}_2\bar{\boldsymbol{x}}_2)^T(\boldsymbol{W}_2\sigma_R(\boldsymbol{W}_1\boldsymbol{p})) + (\boldsymbol{W}_1\bar{\boldsymbol{x}}_1)^T(\boldsymbol{W}_1\boldsymbol{p}). \tag{8}$$

Again, we leverage the assumption that the data mean projects more to the first singular vector of the weight matrix and with Lemma 1, the problem becomes the *dilation* of the second layer,

$$\max_{\boldsymbol{p}:|\boldsymbol{p}|=c} |\boldsymbol{W}_2\sigma_R(\boldsymbol{W}_1\boldsymbol{p})|^2 + |\boldsymbol{W}_1\boldsymbol{p}|^2. \tag{9}$$

We extend the same arguments to further layers and see that the *dilation*s tends to make the network layers approximately additive with respect to the generated perturbation vector. For the last layer, the *dilation* terms are added to objective 4 to account for the errors introduced due to linear approximation of all the ReLU layers. Hence, the final optimization problem for UAP generation becomes,

$$\max_{\boldsymbol{p}:|\boldsymbol{p}|=c} |f_L(\boldsymbol{p})|^2 + \sum_{l=1}^{L-1} |f_l(\boldsymbol{p})|^2, \tag{10}$$

which is clearly a completely data-free formulation.

### 3.3 SEQUENTIAL DILATION ALGORITHM

We leverage the theoretical intuitions from the previous Section to formulate an algorithm for UAP generation in a data-free manner. Note that the direct implementation of optimization 10 through any gradient descent algorithm would lead to sub-optimal solutions as the chances of getting stuck

---

**Algorithm 1:** The sequential dilation algorithm for crafting data-free UAPs. The input is the multi-layer neural network $f$ and the perturbation strength $c$. A set of adversarial perturbations $\{\boldsymbol{p}_l\}_{l=1}^{L}$, one for each layer, is returned as the output. Note that $\lambda$ is the learning rate.

---

$\boldsymbol{p}_0 \sim \mathcal{U}(-10, 10)$
**for** $l = \{1, 2, \ldots, L\}$ **do**
$\quad \boldsymbol{p}_l = \boldsymbol{p}_{l-1}$
$\quad$ **while** *convergence* **do**
$\quad\quad \boldsymbol{p}_l = \boldsymbol{p}_l + \lambda \nabla_{\boldsymbol{p}_l} \sum_{i=1}^{l} \log(|f_i(\boldsymbol{p}_l)|^2)$
$\quad\quad$ Set $|\boldsymbol{p}_l|_\infty = c$
$\quad$ **end**
**end**

---

in local minimas is high. This is especially true since no data is used and the only variable being optimized is $\boldsymbol{p}$ with no sources of randomness. Hence, we perform the *dilations* of optimization 10 in sequential manner so as to avoid chances of reaching local minima solutions. Some more changes are applied in the way the original optimization is implemented, mainly for training stability and to compare fairly with existing methods. For numerical stability of the optimization, we follow Mopuri et al. (2018) and maximize logarithm of the Euclidean norm in the dilate loss. In order to compare with existing methods, $l_\infty$ norm is restricted instead of the $l_2$ in the problem 10. This constrains the maximum of absolute value of the adversarial noise.

Algorithm 1 elucidates our proposed sequential dilation algorithm for ReLU based neural networks. The procedure loops over all the layers of the network. For the first layer, we find a vector $\boldsymbol{p}_1$ which maximizes the logarithm of $l_2$ norm of $\boldsymbol{W}_1 \boldsymbol{p}_1$, essentially finding the first singular vector of $\boldsymbol{W}_1$. After the *dilation* of the first layer, the perturbation $\boldsymbol{p}_1$ is used as an initialization for maximizing the Euclidean norm of second layer. But note the first loss term $|\boldsymbol{W}_1 \boldsymbol{p}|_2^2$ is still kept in the *dilation* of second layer. This loss formulation tries to maximize the norm of output at the current layer along with all the previous layers that feed into it. In short, *dilation* of $l$th layer starts the optimization with perturbation obtained from *dilation* of $(l-1)$th layer and involves the joint *dilation* of all $l$ layers. The method runs till the softmax layer of the network and the final perturbation $\boldsymbol{p}_L$ is a UAP, created without using any training data and could potentially fool majority of input samples.

We only consider CNNs trained for classification task. The optimization is performed using standard ADAM optimizer (Kingma & Ba, 2014) with a fixed learning rate schedule till the training loss saturates. Typical learning rate is $0.1$. At every step of the optimization, the values of the perturbation are clipped to limit the allowed range. The $l_\infty$ norm is set as 10 for all our experiments. Although, Euclidean and maximum norms are not theoretically equivalent, practically we observe that the final perturbations are saturated, with roughly more than 78% of the values reaching $\pm 10$. This implies the $l_2$ norm also to be approximately restricted under the saturation assumption. Once the perturbation gets saturated while optimization, the loss might saturate and could be stuck in local minimas.

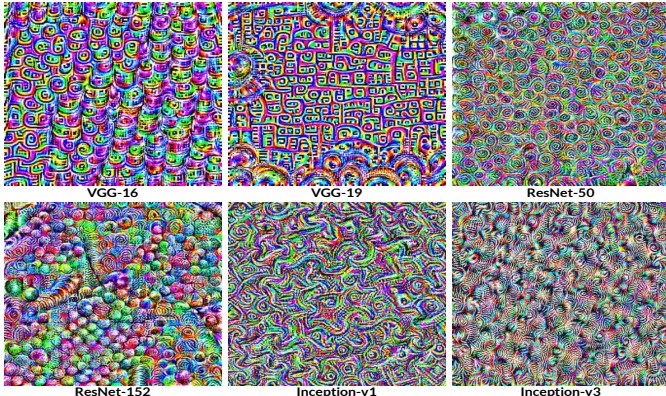

Figure 2: Data-free Universal Adversaries crafted using our sequential dilation algorithm.

Table 1: Comparison of fooling rates (in percentage) obtained for our approach with that of other works. We achieve higher fooling rate than the existing data-free method GDUAP (Mopuri et al., 2018). Note that UAP (Moosavi-Dezfooli et al., 2017) is data dependent (shown for completeness).

| Network | UAP | Random | GDUAP | Ours |
|---|---|---|---|---|
| VGG16 | 77.80 | 8.20 | 44.79 | **51.35** $\pm$ 1.16 |
| VGG19 | 80.80 | 8.14 | 40.51 | **49.58** $\pm$ 0.96 |
| ResNet 50 | - | 10.01 | 32.81 | **60.10** $\pm$ 0.63 |
| ResNet 152 | 84.00 | 8.46 | 23.90 | **47.37** $\pm$ 1.50 |
| Inception v1 | 78.50 | 9.94 | 17.59 | **52.02** $\pm$ 1.05 |
| Inception v3 | - | 7.83 | 21.74 | **47.18** $\pm$ 1.36 |

To prevent this, after dilation at every layer, we rescale the perturbation by dividing the pixel values by 2. This does not make any difference to the procedure as only the magnitude is changed to make room for further optimization. Ideally, we should do sequential dilations for all the convolutional and fully connected layers of the CNN from input side to the end softmax classifier. But for very deep models like Inception and ResNet, the dilations are done only for every architectural blocks. Because of this the optimization might be more nonlinear than what is being assumed in Section 3.1, with the maximum norm further inducing the clipping nonlinearity. Hence, we initialize the perturbation ($\boldsymbol{p}_0$) with the values drawn randomly from a uniform distribution $\mathcal{U}(-10, 10)$. Finally, note that absolutely no data in any form is used for creating adversarial noise, not even a validation set is employed in contrast to Mopuri et al. (2018). The code for our approach is available for review at the anonymous link https://github.com/anoniclr/uap_seq_dilate.

## 4 EXPERIMENTS

We benchmark our proposed sequential dilation method with the existing data-free approaches. All the experiments are performed on popular classification models like VGG (Simonyan & Zisserman, 2014), ResNet (He et al., 2016) and Inception (Szegedy et al., 2016). These models are already trained on Imagenet (Deng et al., 2009) dataset and delivers very high classification accuracy. Figure 2 shows the perturbations crafted using the proposed method for various networks. We follow other works (Moosavi-Dezfooli et al., 2017; Mopuri et al., 2018) and assess the performance of our adversarial attack using the fooling rate metric. Fooling rate is defined as the fraction of test images on which the network prediction differs before and after the addition of the adversarial noise. Table 1 reports the fooling rates obtained by our method along with that of other works. The first comparison is with the random baseline, which is the fooling incurred with just random noise. Second baseline is with the only existing data-free approach GDUAP. Clearly, our proposed data-free objective achieves significantly higher fooling rates than the other data-free work. This indicates that sequential dilation algorithm, not only has theoretical backing, but also results in higher fooling rates in practice. Note that we have run our method ten times and the results produced in the table are mean fooling rate along with the standard deviation, to statistically validate the performance improvement.

Now we ablate the different aspects of the sequential dilation algorithm to demonstrate the usefulness of the design choices. Table 2 reports the results of the various ablative experiments. First

Table 2: Various ablative experiments regarding our approach. It is evident that the proposed sequential dilation algorithm has higher performance than any of its variants.

| Network | Single Dilation | PSM Maximization | Ours without accumulation | Ours |
|---|---|---|---|---|
| VGG16 | 49.46 | 24.24 | 31.49 | **51.35** |
| VGG19 | 44.10 | 25.20 | 31.00 | **49.58** |
| ResNet 50 | 25.62 | 16.57 | 28.52 | **60.10** |
| ResNet 152 | 28.16 | 15.85 | 13.40 | **47.37** |
| Inception v1 | 24.81 | 15.57 | 39.08 | **52.02** |
| Inception v3 | 19.40 | 11.92 | 26.08 | **47.18** |

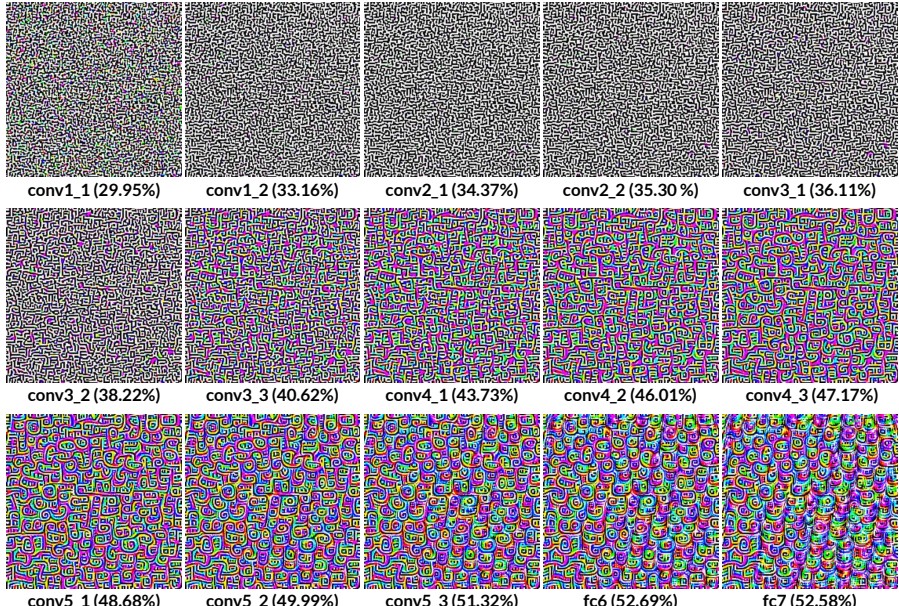

Figure 3: Intermediate perturbations ($p_l$s) crafted for different layers using our sequential dilation algorithm for VGG-16 network with corresponding fooling rates obtained.

experiment is non-sequential version of dilation, listed as *single dilation* in the table. This is a single joint optimization maximizing the norm of activations before the non-linearity. *PSM maximization* refers to simple maximization of the pre-softmax layer ($f_L$) alone, which is same as the approximated objective 4. As described in Section 3.3, our dilation of a layer involves keeping the maximization terms of all the previous layers. We empirically validate the necessity of such a scheme by just sequentially maximizing the layer norms without the cumulative loss term for the experiment *Ours without accumulation* in Table 2. Note that each maximization starts with the perturbation initialized from the previous optimization. The results for these ablations evidence that our exact formulation of sequential dilation achieves higher fooling rates in data-free scenario. Further, Figure 3 displays the perturbations obtained through sequential dilation at every layer for VGG-16 network, basically the $p_l$s from Algorithm 1. We also indicate the corresponding fooling rates for each of the perturbation. It is interesting to observe that the fooling rate increases as we successively dilate layers and saturates towards the end, again emphasizing the need for the sequential process.

In many practical attack scenarios, the actual deployed model might not be available to generate the adversarial perturbation. Hence, the ability of the perturbation crafted for one network to cause reasonable fooling on another network is often highly sought-after property. This setting is known as black-box, for which we compare our method with GDUAP (Mopuri et al., 2018) in Table 3. The results evidence better black-box performance for our method than the existing data-free work, suggesting higher generalization for the perturbations from sequential dilation.

Table 3: Black-box testing of data-free sequential dilation approach. Rows indicate the networks used to craft the adversary and the columns specify the one for which the fooling rate is computed. Blue values are the fooling rates obtained by our method, while the red ones correspond to GDUAP.

| Network | VGG16 | VGG19 | ResNet 50 | ResNet 152 | Inception v1 | Inception v3 |
|---|---|---|---|---|---|---|
| VGG16 | 44.8, 52.2 | 38.5, 45.9 | 27.7, 37.8 | 23.9, 32.1 | 26.3, 34.9 | 20.1, 26.9 |
| VGG19 | 35.6, 47.5 | 40.5, 52.0 | 24.9, 37.3 | 21.5, 30.4 | 25.7 , 33.7 | 18.7, 27.2 |
| ResNet 50 | 15.2, 23.6 | 14.8, 23.1 | 32.8, 59.8 | 17.9, 35.0 | 18.9, 33.3 | 13.7, 21.6 |
| ResNet 152 | 16.6, 27.1 | 15.8, 26.3 | 21.7, 36.6 | 23.9, 46.7 | 18.7, 27.5 | 13.7, 18.5 |
| Inception v1 | 14.4, 26.1 | 13.9, 26.5 | 16.4, 34.3 | 14.1, 27.2 | 17.6, 51.5 | 12.1, 21.8 |
| Inception v3 | 11.8, 16.5 | 11.8, 16.3 | 13.2, 18.0 | 11.8, 14.6 | 13.9, 16.5 | 21.7, 45.9 |

Table 4: Comparison of fooling rates obtained by our method and GDUAP (Mopuri et al., 2018) in less data setting with varied number of images available for crafting the perturbation.

| Network | GDUAP | | | | Ours | | | |
|---|---|---|---|---|---|---|---|---|
| | D=64 | D = 500 | D=1000 | D=10000 | D=64 | D = 500 | D=1000 | D=10000 |
| VGG16 | 71.25 | 75.16 | 76.20 | 75.62 | 74.25 | 77.66 | 78.12 | 79.09 |
| VGG19 | 68.65 | 73.11 | 74.19 | 74.73 | 66.81 | 72.34 | 74.00 | 74.28 |
| ResNet 50 | 57.94 | 71.98 | 72.71 | 71.64 | 70.33 | 74.52 | 74.54 | 76.36 |
| ResNet 152 | 43.95 | 67.43 | 69.34 | 70.48 | 68.07 | 70.57 | 71.52 | 71.64 |
| Inception v1 | 15.83 | 85.99 | 86.41 | 87.10 | 60.26 | 64.43 | 69.68 | 71.78 |
| Inception v3 | 27.13 | 53.84 | 55.04 | 59.88 | 81.54 | 86.69 | 87.21 | 87.57 |

Table 5: Benchmarking our approach in less data scenario with Singular Fool (Khrulkov & Oseledets, 2018) where the adversaries are crafted with just 64 images.

| Network | Ours | | Singular Fool |
|---|---|---|---|
| | Imagenet | Pascal | (Khrulkov & Oseledets, 2018) |
| VGG16 | 72.13 | 76.15 | 52 |
| VGG19 | 66.56 | 68.87 | 60 |
| ResNet 50 | 68.73 | 65.69 | 44 |
| ResNet 152 | 66.84 | 66.47 | - |
| Inception v1 | 46.74 | 40.86 | - |
| Inception v3 | 76.42 | 76.70 | - |

The experiments performed so far show that proposed sequential dilate loss formulation achieves state-of-the-art fooling rates in data-free scenarios. We now consider the case where minimal training data is available, called the less data setting. For this case, the sequential dilation is applied with the limited data. The input to the network at any stage of the optimization is the image added with the current perturbation ($x_i + p_l$ for layer $l$). With the help of some data points, we expect the solution to approach more closer to the actual adversarial perturbation obtained with full data. Table 4 indicates the fooling rates of the less data setting with varied amount of training samples. Note that to compare with GDUAP, we also use a validation set to select the best perturbation while training. Our approach performs significantly better than GDUAP when data samples are very less, increasing the practical utility of the method. We also observe that the fooling rates with less data, in general, have increased than data-free and became comparable to full data UAP (see Table 1).

Furthermore, Table 5 compares our approach with Singular Fool (Khrulkov & Oseledets, 2018) in extremely less data scenario. For fair comparison with Khrulkov & Oseledets (2018), we use only 64 images for crafting the perturbation and no validation set is employed. The best perturbation is selected based on the training loss. As expected, our method achieves significantly higher fooling performance than Khrulkov & Oseledets (2018). Even more, we apply our algorithm with 64 randomly chosen images from Pascal VOC (Everingham et al., 2011). Interestingly, despite the models used are being trained for a different dataset, the fooling rates remain more or less similar and is higher than that of Khrulkov & Oseledets (2018). This shows that our approach works well in less data cases even when the available images are not from the dataset on which the model is trained.

## 5 CONCLUSIONS AND FUTURE WORK

In this paper, we have presented a new algorithm, called the sequential dilation, to craft universal adversaries in a data-free manner. The approach relies on finding the first singular vector of the linearly approximated neural network. The approximation is being enabled by optimizing with the proposed *dilate* loss. Elaborate experiments and ablations demonstrate that our approach achieves superior data-free fooling performance. One promising direction for future research would be to modify the algorithm and generate targeted UAP, where the objective is fooling to a specific class.

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

## A  PROOF OF LEMMA 1

If $W = USV^T$ denote the singular value decomposition of $W$, then

$$\max_{p:|p|=c} x^T W^T W p = x^T V S^2 V^T p = \sum_i S_{ii}^2 (x^T V_{:,i})(p^T V_{:,i}). \tag{11}$$

Since $x^T V_{:,1} > x^T V_{:,i}$ (by assumption) and $S_{11} > S_{ii}$ for all $i > 1$, the solution to the optimization is $cV_{:,1}$, a scaled version of the first singular vector of $W$. The same solution can be obtained through the definition of the first singular vector as,

$$\max_{p:|p|=c} |Wp|^2, \tag{12}$$

completing the proof.

## B  EXPERIMENTAL SETUP

All our experiments reported in the main paper are run on NVIDIA DGX cluster (Dual 20-core Intel Xeon E5-2698 v4 2.2 GHz) within the Tensorflow docker. We use the pretrained classification models from Tensorflow Slim library S. Guadarrama (2016).

## C  DEMONSTRATION OF ADVERSARIAL ATTACK

Figures 4 to 9 demonstrate adversarial attack through perturbation generated from our proposed sequential dilation algorithm for various networks. The first row in the figures shows the clean images with the class predicted by the model, while the second row has the corresponding perturbed samples with the flipped prediction labels.

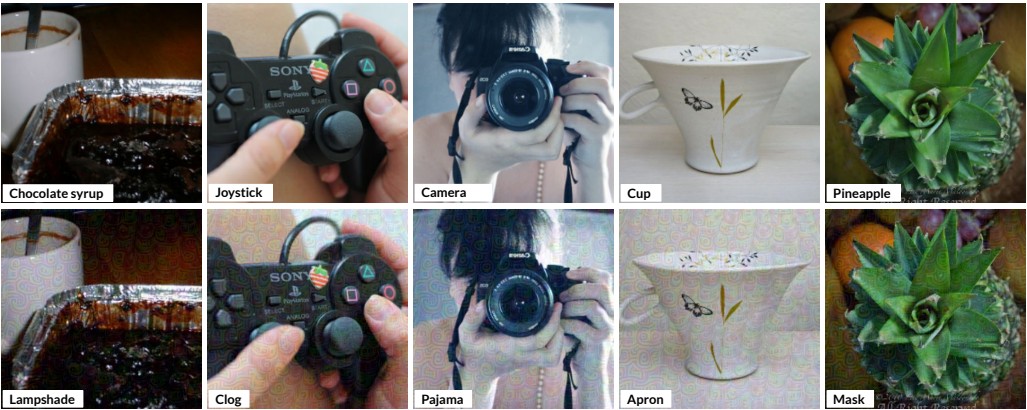

Figure 4: Attack on VGG-16 network.

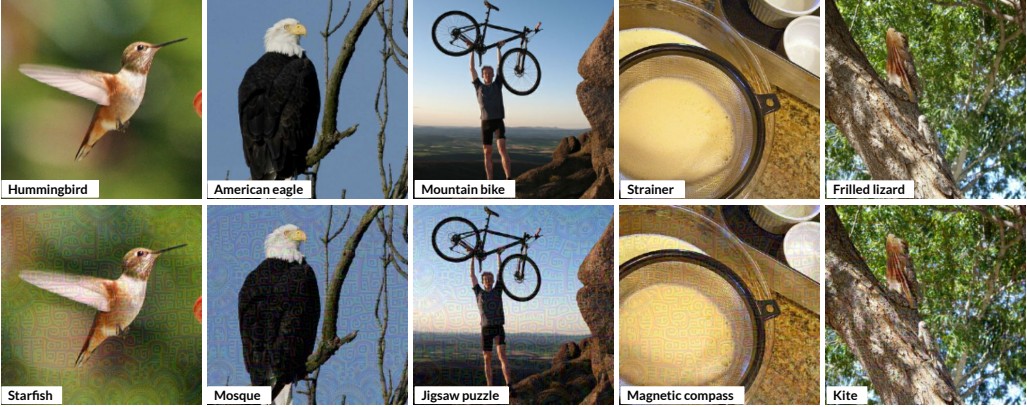

Figure 5: Attack on VGG-19 network.

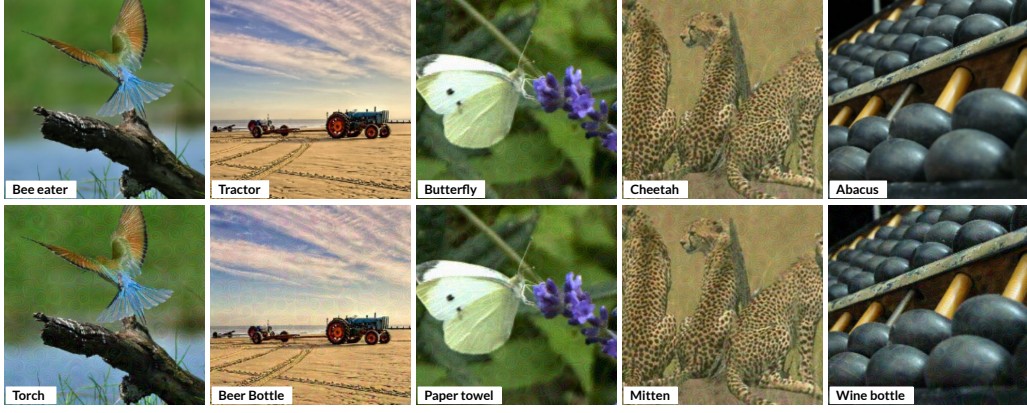

Figure 6: Attack on ResNet-50 network.

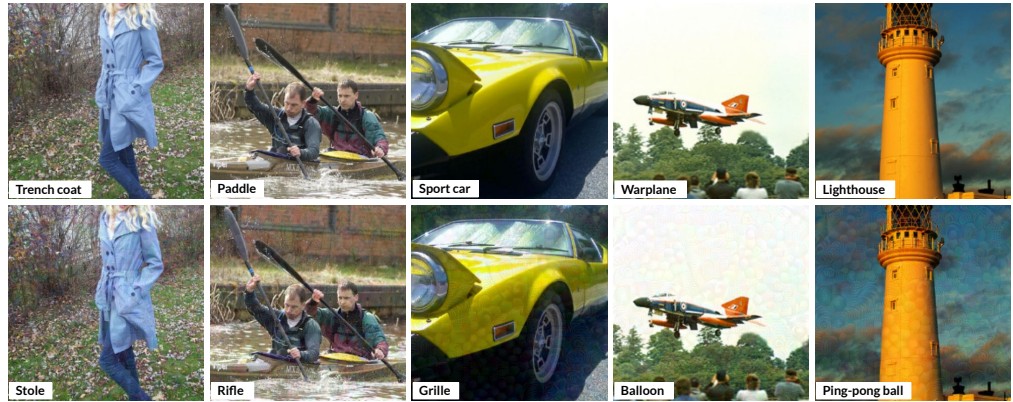

Figure 7: Attack on ResNet-152 network.

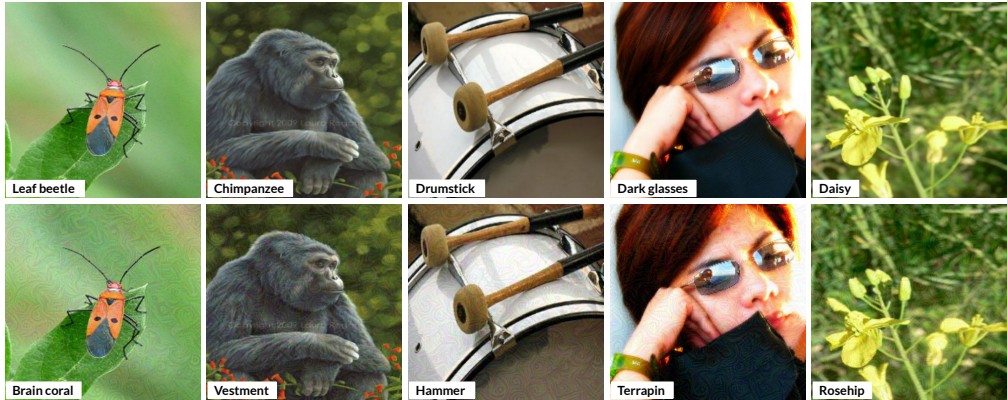

Figure 8: Attack on Inception-v1 network.

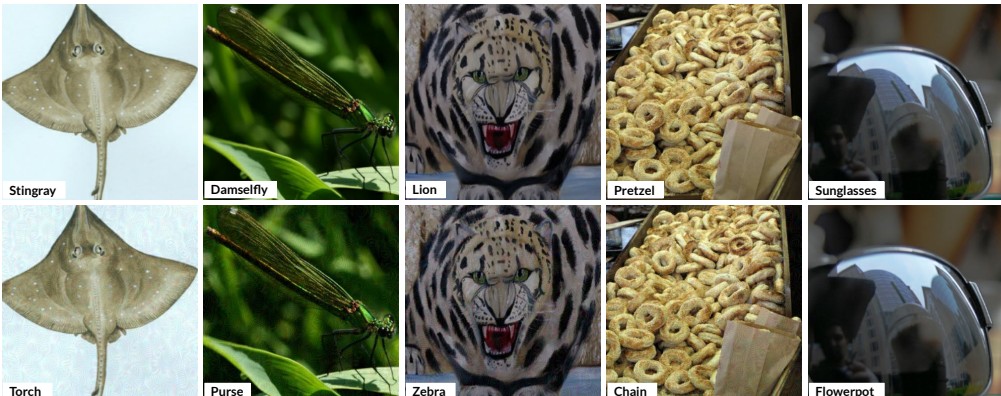

Figure 9: Attack on Inception-v3 network.

