# OpenReview forum: "Crafting Data-free Universal Adversaries with Dilate Loss"
_ICLR.cc/2020/Conference — Reject_

### Official Review · AnonReviewer1 · 2019-10-22
**Official Blind Review #1**

**Rating:** 8

**Review:**

The paper is well written and easy to follow. In this paper, a new data-free method is proposed to create universal adversarial perturbation without using data. There are some similarities with GDUAP though, authors also make some crucial improvements. They perform Euclidean norm maximization before the non-linearity of each layer, which not only has theoretical backing but also brings better performance in practice. Meanwhile, they optimize the perturbations in each layer in a sequential manner instead of joint optimization to avoid chances of reaching local minima solutions.

The authors provide a detailed theoretical analysis and systematic experimental results to demonstrate their arguments, which is convincing. What’s more, the proposed method achieves state-of-the-art data-free fooling rates on the large-scale dataset, which strongly demonstrates the effectiveness of their method.

In section 3.2, (the top of page 4) “which becomes additive if column vectors in W1X are in the same orthant as W1p. We relax this criteria and favour the case of making the vectors as close as possible by”
Could the authors provide more discussions about it?

**Experience Assessment:**

I have read many papers in this area.

**Review Assessment: Checking Correctness Of Derivations And Theory:**

I assessed the sensibility of the derivations and theory.

**Review Assessment: Checking Correctness Of Experiments:**

I assessed the sensibility of the experiments.

**Review Assessment: Thoroughness In Paper Reading:**

I read the paper at least twice and used my best judgement in assessing the paper.

---

> ### Author Response · Authors · 2019-11-14
> **Response to Reviewer 1**
>
> We thank the Reviewer for the valuable feedback.
>
> 1. Section 3.2 (the top of page 4) clarification: Additive means that $\sigma_{R}(W_{1}X+W_{1}p1^{T})=\sigma_{R}(W_{1}X)+\sigma_{R}(W_{1}p1^{T})$. Please see the response to Reviewer 4 Q1.

---

### Official Review · AnonReviewer2 · 2019-10-23
**Official Blind Review #2**

**Rating:** 3

**Review:**

This paper proposed a white-box (known network architecture, known network weight) data free (without need to access the data) adversarial attacking method. The main idea is to find a perturbation that maximizes the activations at different layers jointly. But the optimization is done sequentially, treating each layer’s activation (before ReLU) as a linear transformation output.

The method is compared with existing methods (only one existing approach for the problem, GDUAP by Mopuri et al. 2018) in terms of the fool rate. It shows significant improvement. Ablation study is carried out to compare with baselines like perturbation maximizing only first layer activation, only last layer activation, etc. Also on some other settings (black-box testing, less data) the proposed method outperforms GDUAP.

The problem of data-free white-box attack is very interesting and does make sense. The proposed method achieve significant improvement over the previous one (GDUAP). I do have the following concerns though.

1), the novelty of the proposed idea seems relatively limited. The proposed idea seeks perturbation maximizing activations over all layers. It incur perturbation before ReLU. But overall, the flavor of the idea is not significantly different from GDUAP, despite the significant performance boost.

2), it was mentioned that compare with GDUAP, this paper has more theoretical analysis. But this is not very convincing to me. There are many steps of approximation/relaxation from the original problem (Equation (1)) to the final formula (Equation (10)). Many assumptions are made over the steps. It is OK to use these steps to derive a heuristic. But these steps can hardly be called "theoretical analysis".

I am particularly uncomfortable with Equation (5), which is the basis of the main idea. It assumes that all data in $W_1X$ are in the same orthant as $W_1p$. But this is unrealistic as different data in X will for sure incur different activation patterns. Did I misunderstand anything?

3) I do like the experimental results. It looks impressive. But the baselines are really limited (granted, there are not many existing approaches). There is only one task (image classification). How about other tasks like segmentation etc shown in Mopuri et al. 2018? Also it would be nice to also show the results of other UAP methods, as it gives us a better sense of the gap between with and without data.

4) I wonder how will the attack affect some model which has been trained with some defense mechanism, e.g., adversarial training.

Typo:
Equation (5), RHS missing a max



**Experience Assessment:**

I do not know much about this area.

**Review Assessment: Checking Correctness Of Derivations And Theory:**

N/A

**Review Assessment: Checking Correctness Of Experiments:**

I assessed the sensibility of the experiments.

**Review Assessment: Thoroughness In Paper Reading:**

I read the paper at least twice and used my best judgement in assessing the paper.

---

> ### Author Response · Authors · 2019-11-14
> **Response to Reviewer 2**
>
> We thank the Reviewer for the valuable feedback.
>
> 1. Novelty: Though our method has similar flavor of GDUAP, as mentioned by Reviewer 1, our approach has significant improvements over GDUAP: (1) our work formulates the proposed method from theoretical intuitions (2) provides better understanding than purely empirical method in GDUAP, which could pave way for further studies in the field (3) achieves significantly high fooling rate than GDUAP.
>
> 2. Theoretical analysis and Equation (5): As pointed by the reviewer, we arrive at the proposed method by intuitions derived from a set of reformulations and approximations. We believe such a scheme also could be considered an analysis as it explains why the method works.  Regarding equation (5), please see the response to Reviewer 4 Q1.
>
> 3. Results on other tasks: We implement the proposed algorithm on FCN-8s-VGG model (please refer to GDUAP paper) for segmentation task as suggested by the Reviewer. Our method is able to bring down the mIoU after attack better than GDUAP, indicating superior adversarial performance.
> ------------------------
> Method   | mIoU
> ------------------------
> Original | 65.49
> GDUAP    | 42.78
> Ours     | 36.58
> ------------------------
>
> 4. Adversarial trained model: We are able to test our algorithm on adversarial trained Inception V3 model (from "Tramer et al. Ensemble Adversarial Training: Attacks and Defenses [ICLR 2018]", available at https://github.com/tensorflow/models/tree/master/research/adv_imagenet_models) and achieve high fooling rate reported as below:
> ------------------------
> Method   | Fooling Rate
> ------------------------
> GDUAP    | 33.33%
> Ours     | 59.14%
> ------------------------
> This clearly demonstrates the effectiveness of the proposed sequential dilation algorithm.
>
> 5. Typo: We will correct the typo.

---

### Official Review · AnonReviewer3 · 2019-11-01
**Official Blind Review #3**

**Rating:** 6

**Review:**

The paper proposes a data free method for generating universal adversarial examples. Their method finds an input that maximizes the output of each layer by maximizing the dilation loss. They gave a well motivated derivation going from the data matrix, the data mean and to data free. The experiments results seems solid as the numbers show that their method is much better in many cases.

I have 2 main issues:

* The fooling rate experiments does not seem to control for how much distortion there really is. How do you make sure that different methods have similar level of distortion and not just similar l_\inf.  Given that the authors says most of their method saturates all values, it is not clear that the baselines and competition really has a similar level of distortion. The fooling rate for random seems rather high. Why is random noise not mostly ignored by the model?

* while the method is data free. It needs complete access to the model and relies on properties of ReLu. I am not sure how realistic this setting is, and how this compares to methods that has black box access to the model. While it is interesting, the paper did not establish that universal adversarial perturbation is well-motivated and why data free is more important that model free or targeted perturbations.  An attacker probably always see the input and probably wants to make it misclassified into a particular class, instead of just making the model wrong.



**Experience Assessment:**

I do not know much about this area.

**Review Assessment: Checking Correctness Of Derivations And Theory:**

I assessed the sensibility of the derivations and theory.

**Review Assessment: Checking Correctness Of Experiments:**

I assessed the sensibility of the experiments.

**Review Assessment: Thoroughness In Paper Reading:**

I made a quick assessment of this paper.

---

> ### Author Response · Authors · 2019-11-14
> **Response to Reviewer 3**
>
> We thank the Reviewer for the valuable feedback.
>
> 1. Distorton rate: L-infinity norm is the criteria widely used in the community and all comparisons are based on this. Nevertheless, we checked and found that GDUAP has similar saturation/distortion as our method. In Table 1, random accuracies are below 10% and this might be due to noise affecting some class discriminative regions of the images, especially for those images falling near the decision boundaries.
>
> 2. Practicality of data-free approach: In many practical scenarios, the model might be made available, but the training data cannot be released due to privacy/confidentiality reasons (e.g., medical records, proprietary data). Data-free approaches suite such cases and in fact Table 3 shows good black-box performance (better than GDUAP) for our method, increasing its practical utility.

---

### Official Review · AnonReviewer4 · 2019-11-03
**Official Blind Review #4**

**Rating:** 3

**Review:**

Summary:
This paper proposed a method to generate universal adversarial perturbations without training data. This task is timely and practical. The proposed method maximizes the norm of the output before nonlinearity at any layer to craft the universal perturbation. A sequential dilation algorithm is designed to calculate UAPs. The experiments show that the proposed method outperforms GDUAP.

My major concern is that there is not much novelty in the proposed method compared with GDUAP. The dilate loss function (4) is similar to the objective function (3) in the GDUAP paper. This paper provides a theoretical explanation of the dilate loss function and an improvement on the non-linearity function, which, however, is not convincing. Equation 10 is derived based on many strong assumptions. See the comments below.

Pros:
-	The theoretical analysis is clear.
-	The proposed method performs better than GDUAP in the data-free and black-box setting.
-	The writing is good. The paper is easy to follow.

Cons:
-	The theoretical analysis is based on many strong assumptions/criteria. For example:
o	To derive equation (5), W1X and W1p must be in the same orthant. It is unclear how to satisfy the criteria In the algorithm.
o	In Lemma 1, problem (5) approximates problem (6) only if x has a very large projection on the first singular vector of W. However, x and W are fixed and independent of p. This assumption largely depends on the dataset and the weights of the model.
o	It would be better if the authors show that in what cases these assumptions can be satisfied.
-	Other factors such as batch normalization and max pooling used in Inception v3, may also affect the linearity of the model. It would be better if the authors provide theoretical analysis or an ablation study on these factors.
-	What’s the design principle behind Algorithm 1? Why can this algorithm solve the sub-optimal problem? The weights of different layers are not closely related. In the initialization part, why can we start learning p from the result of the previous layer? Would it be possible that the performance is improved due to the algorithm instead of the dilate loss?
-	The proposed method performs worse than GDUAP does in some less data settings.
-	The results in Table 4 and 5 are inconsistent. These two experiments use the same dataset (Imagenet) and the same number of images (D=64).



**Experience Assessment:**

I have published one or two papers in this area.

**Review Assessment: Checking Correctness Of Derivations And Theory:**

I carefully checked the derivations and theory.

**Review Assessment: Checking Correctness Of Experiments:**

I assessed the sensibility of the experiments.

**Review Assessment: Thoroughness In Paper Reading:**

I read the paper at least twice and used my best judgement in assessing the paper.

---

> ### Author Response · Authors · 2019-11-14
> **Response to Reviewer 4**
>
> We thank the Reviewer for the valuable feedback.
>
> 1. Reg. Equation (5): To derive equation (5), we do not require $W_{1}p$ to be in the same orthant as column vectors of $W_{1}X$. In fact, such a $p$ almost never exists. Hence, our approach is to find a $p$ that can minimize the error due to the additive approximation of ReLU ($\sigma_{R}(W_{1}X+W_{1}p1^{T})\approx\sigma_{R}(W_{1}X)+\sigma_{R}(W_{1}p1^{T})$). We linearly relax this criteria and search for a $p$ that can bring $W_{1}p$ as close to all the column vectors of $W_{1}X$. This is realized through optimization (5) by maximizing the inner product of $W_{1}p$ with all the column vectors of $W_{1}X$.
>
> 2. Assumption for Lemma 1: Since our method is data-free, there must be an assumption tying data to the learned weights of the network. We assume that the singular vectors of the weights must have captured the discriminatory modes of data samples while training. This means that the first singular vector carries the most important features common to most of the data points than the other singular vectors, which translates to the assumption for Lemma 1. This assumption seems to be required for explaining the high fooling rate our method obtains.
>
> 3. Other nonlinearities in the network: Currently, our theoretical explanation is limited to ReLU nonlinearity, which itself seems sufficient to reason out the high fooling performance. The effect of other kinds of nonlinearities needs to be studied in future works.
>
> 4. Design of Algorithm 1: Algorithm 1 is designed from problem (10), where it is implemented as a set of sequential optimizations. We have an ablation in Table 2 with the header 'Ours without accumulation', where we optimize without the 'dilate' loss exactly like as the Reviewer mentioned. The results clearly evidence the performance boost with the 'dilate' loss.
>
> 5. Results in less data cases: For VGG19 and Inception v1, with more data there is a slight dip in fooling rate. This seems to be something specific to the networks, but for majority cases the proposed approach beats GDUAP.
>
> 6. Results in Table 4 and 5: For Table 4 experiments, a validation set is used to select the best perturbation. But in Table 5, in order to compare with Singular Fool method, we do not employ a validation set as mentioned in the text. The slight difference in fooling rate is due to this fact.

---

### Decision · Program_Chairs · 2019-12-19

**Decision:**

Reject

**Comment:**

This paper focuses on finding universal adversarial perturbations, that is, a single noise pattern that can be applied to any input to fool the network in many cases. Further more, it focuses on the data-free setting, where such a perturbation is found without having access to data (images) from the distribution that train- and test data comes from.

The reviewers were very conflicted about this paper. Among others, the strong experimental results and the clarity of writing and analysis were praised. However, there was also criticism of the amount of novelty compared to GDUAP, on the strong assumptions needed (potentially limiting the applicability), and on some weakness in the theoretical analysis.

In the end, the paper seems in current form not convincing enough for me to recommend acceptance for ICLR.